# Open Data Fusion for Approximate True Cost Accounting using Knowledge Graphs

Bileam Scheuvens[1,*], Harrisen Scells[1]

[1]*University of Tübingen, Tübingen, Germany*

### Abstract

Economic theory describes how price mechanisms aggregate signals such as supply and demand in markets to efficiently allocate resources. Reflecting negative externalities such as pollution in market prices is necessary for long-term price stability. In practice the process of modelling externalities is difficult and time consuming even for individual products, which prevents large scale adoption and leads to valuable signal being ignored. We aim to answer the question of whether the process of estimating a "true price" which includes externalities can be automated and propose a framework that aggregates existing data from different sources into a Knowledge Graph (KG). Our approach streamlines the process of data acquisition and processing by enforcing a high level schema. Our tooling is easily extensible, handles varied data from the aggregation platform Wikirate alongside the OpenProductsFacts database and allows for an approximation of true costs for any company with sufficient data availability. We present a demo system that includes over 900 companies at https://atca.health-nlp.com. The underlying open source code is available at https://github.com/bileamScheuvens/HiddenCostReport.

### Keywords

True Cost Accounting, Knowledge Graphs, Data Fusion, TEXT2SPARQL, QLever

## 1. Introduction and Related Work

Disregarding harm which affects third parties, such as pollution or worker exploitation in economic transactions, leads to prices settling below the net value the transactions contribute to society [1]. These false price equilibria reduce market efficiency and incentivize unsustainable practices by effectively subsidizing actions with hard-to-assess consequences. Market regulation via prohibition can address these issues to an extent, but legislation is often slow to adapt, imprecise, and limited to regional authority [2]. Instead, one can apply a pigouvian tax [1] to a product by assigning costs to all externalities that cover restoration and compensation of the affected parties. This enables price finding mechanisms to restore market efficiency as the previously violated assumption that all relevant costs and benefits are considered is now satisfied. The difficulty in applying such a framework in practice lies in accurately assessing and assigning costs to externalities, a practice termed True Cost Accounting (TCA). We propose an approach that uses a KG to represent companies and externalities to streamline the process of aggregating hidden costs to estimate a true cost for arbitrary products using publicly available data.

Several works exist which explore the concept of pricing displaced harm with varying methodologies and target audiences. The True Price Foundation standardized and open-sourced a method for calculating a true price,[1] which has been applied to different products [3, 4, 5, 6]. TEEBAgriFood [7] is a division of the The Economics of Ecosystems and Biodiversity (TEEB) initiative, which itself is part of the United Nations Environment Programme (UNEP). It currently spans 14 countries and prices natural factors that are usually ignored in an attempt to influence policy making. The TCA Accelerator is a network that advocates for the use of TCA together with the Global Alliance for the Future of Food,[2,3] focusing on interdisciplinary knowledge sharing.

---

*NORA'26: Workshop on KNOwledge GRaphs & Agentic Systems Interplay*

*Corresponding author.

✉ bileam.scheuvens@student.uni-tuebingen.de (B. Scheuvens); harrisen.scells@uni.tuebingen.de (H. Scells)

🆔 0009-0006-5632-9421 (B. Scheuvens); 0000-0001-9578-7157 (H. Scells)

[1]https://www.trueprice.org/about-us/

[2]https://actionagenda.tcaaccelerator.org/

[3]https://futureoffood.org/

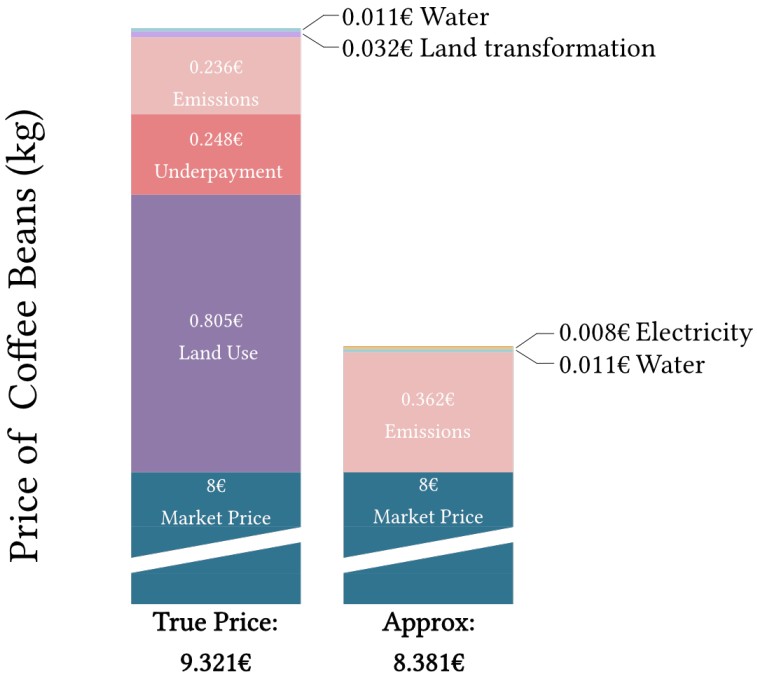

**Figure 1:** Approximate True Cost Accounting (ATCA) result for Bocca Coffee Beans (right) compared to real data from [6](left).

These initiatives tackle the core problem of pricing externalities, but do not scale without significant effort. Other related work such as Davis and Caldeira [8] or Hong et al. [9] may be summarized as Environmentally Extended Input-Output Analysis (EEIOA) which rather model global supply chains as a whole. They include computational models and scale well but crucially avoid the step of converting consumption to cost equivalent, instead simply describing the raw resource production and consumption.

To the best of our knowledge, no prior work addresses the problem of automating the process of TCA end-to-end without needing to gather additional data. Specifically for true pricing the existing efforts also disproportionally focus on agriculture, while our approach intends to work regardless of industry.

## 2. True Cost Accounting using Knowledge Graphs

Traditional TCA inspects the lifecycle of a product to find externalities not reflected by the price. It assumes unrestricted access to company data and supply chain introspection, which are almost never publicly accessible. To illustrate how we may approximate TCA with public data, we may try to reproduce an existing report about the true cost of coffee beans for a particular roaster such as "Bocca Coffee" [6]. We may know the revenue in a given year to be €6.9 million. For illustrative reasons, imagine we measure emissions to be 100 tons of $CO_2$ equivalent alongside water consumption of $6000 m^3$ and electricity consumption of 100 Mwh. We may then consult a table of costs for each externality,[4] to find emissions priced at 312 €/ton and water at 1.62 €/$m^3$. For electricity consumption we use the estimate of 70 €/Mwh by Sovacool et al. [10]. Aggregating these priced externalities lets us compute a hidden cost per euro of revenue, which in turn can be multiplied with the price of a product to obtain an approximate true price under the assumption, that all products contribute to all externalities in proportion to their price. Figure 1 illustrates this compared to a thorough TCA report. With plausible data, the approach undershoots the ground truth due to incomplete data. Nonetheless it contains insights, that the market price alone did not capture. The rest of this section explains the architecture we developed to scale this approach.

---

[4]In this case using the monetization table from https://github.com/Truepricemethod/Monetisation_factors see Appendix B

```
PREFIX hcr: <http://hiddencostreport.org/schema#>
PREFIX wdt: <http://www.wikidata.org/prop/direct/>

SELECT ?company_name ?industry WHERE {
  ?company hcr:Name ?company_name ;
           hcr:OpenCorporatesID ?OCID .

  SERVICE <https://query.wikidata.org/sparql> {
    ?wdEntity wdt:P1616 ?OCID ;
              wdt:P452 ?industry .
  }
}
```

**Figure 2:** Example SPARQL query to fetch industries of a company from Wikidata [14] using a federated query.

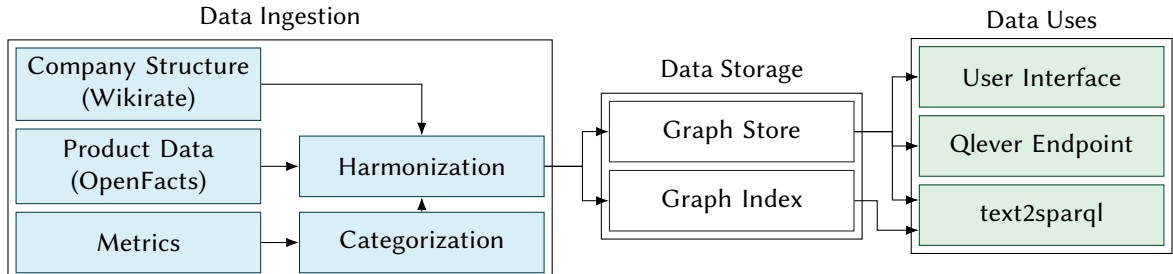

**Figure 3:** A data ingestion component transforms data into a common format followed by on-disk storage and indexing. The data is then consumed by interfaces: the UI, the Qlever endpoint and our text2sparql system.

## 2.1. System Overview

At the core of this demo lies a KG supplemented by a User Interface (UI). Figure 3 provides an overview of the architecture, which is split into three parts: ingestion, storage, and uses, described below.

**Ingestion** The KG is constructed from Wikirate [11] and OpenProductsFacts [12], with sources being toggleable from the UI to suit different use cases; e.g., a targeted analysis of fast food products might rely on OpenFoodfacts [13] for ingredient information, where investigations into child labour might not need it. Wikirate is an open data platform which crowdsources various company data relating to Environmental, Social and Governance (ESG) issues. For the scope of our demo, a subset of 1000 companies and 1000 metrics is transformed into Resource Description Framework (RDF) triples according to the schema shown in Appendix A. At the time of writing, OpenProductsFacts only includes around 40,000 products. As a placeholder for missing products, we allow specifying a price for a product manually. Additional sources may directly be integrated into the graph, or injected at query time through the use of federated queries. The latter allows convenient reuse of existing resources linked through common attributes such as the OpenCorporatesID for companies, illustrated in Figure 2.

All sources are harmonized and disambiguated with a set of heuristics, i.e., applying a form of stemming to company names to strip away organizational indicators such as "inc." or "limited", supplemented by manual curation. Additionally, metrics are categorized to allow unit conversion and cost lookup from a predefined table. Our hierarchical keyword matching approach assigns a category to approximately 30% of metrics, which may already be close to the ceiling, as many metrics such as the number of employees or readability of financial reports cannot meaningfully be translated to costs directly.

To check this assertion we randomly sample 50 metrics and inspect the assigned mapping. We find 17 (37%) correctly assigned metrics, of which five are monetizable, nine are disclosure rates and three are derived metrics, such as *energy consumption per dollar of revenue.* Among the monetizable metrics is *water discharge quality* which has its unit implicitly misinterpreted as tons of water, when in actuality it reports tons of organic material within the water discharge. Of the remaining 33 metrics, three derived metrics and two disclosure rates are missed. None of the unclassified metrics in the sample are easily monetizable.

**Data Storage**  After parsing into RDF triples the data is stored in an Oxigraph [15] database, which uses the RocksDB key-value store internally,[5] chosen for its speed and high-level interface. While parsing we build an index of company names and known aliases mapped to their ID to avoid unnecessary string matching at runtime. Next the data is additionally indexed using Qlever [16] to allow query autocompletion and faster string search. Since Qlever does not allow editing the graph once indexed, we keep the Oxigraph store and only defer to Qlever for exploratory queries.

**Data Uses**  Once the KG is constructed, we provide two ways to interact with it: Firstly ATCA through the UI, where a company and date may be specified, which triggers running pre-written queries and generates a visualization of the approximate hidden cost by metric. We also visualize the evolution of the approximate hidden cost for a window surrounding the queried year to alert to potential outliers from data quality issues.

Secondly, we offer direct query access to the graph for fine grained control. While possible through the UI, we also expose a Qlever [16] endpoint, which provides convenience features such as autocomplete, syntax highlighting and query performance analysis. To improve accessibility to users unfamiliar with SPARQL syntax, we also provide a Large Language Model (LLM)-aided text-to-SPARQL interface.

## 3. Discussion and Conclusion

This work investigated the possibility of estimating a holistic price for any product from available data on externalities caused by the producing company. True pricing is an indispensable tool for policymakers, consumers, and producers alike if we aim to reimagine our global supply chains in a sustainable way. A requirement of this vision is that the tools involved, like that of this demo, are accessible and transparent. The success of platforms such as Wikirate [11] already demonstrates the public demand for accountability, but the nature of individual and disconnected data points makes drawing concrete conclusions difficult. Our demo and its open data KG thus constitute the first concrete steps towards making true cost approximations more accessible. Furthermore, as data becomes increasingly available due to more interconnected and monitored economy, tools such as ours will become much more informative in the near future.

**Limitations and Future Work**  A limitation of the system that remains is the difficulty of evaluation. Besides a handful of true cost reports, which may act as sanity checks, there is no reliable ground truth. One may perform quality control on the KG as a proxy for uncertainty, for example using the methodology of [17] but reliably quantifying the expected error remains difficult.

Additionally, the approach makes strong simplifying assumptions, which are often violated in practice. To name a few: "All products contribute to all externalities in proportion to their market price", "The entire supply chain is vertically integrated" or "The costs associated with metrics are location independent and constant". Loosening these assumptions would be a valuable contribution, e.g., with a cost model as a function of time and space or a unified way of addressing over- or undercounting of externalities along a distributed supply chain.

As future work, we will make the pipeline more robust. The harmonization and categorization of sources rely on heuristics and keyword matching, and remain vulnerable to edge cases. Metrics such as "Reduction of hazardous waste" are commonly categorized under hazardous waste, which skews results. Deduplication is also not trivial, as differences in gathering or preprocessing between metrics that target the same underlying concept make determining authority for competing values challenging. Finally the LLM-based Text2SPARQL interface is error-prone and a more sophisticated approach such as an agentic system with query access is warranted to replace the current zero-shot setup.

---

[5]https://rocksdb.org/

## Acknowledgments

The authors gratefully acknowledge the computing time granted by the KISSKI project. Some calculations for this research were conducted with computing resources under the project HiddenCostReport.

## Declaration on Generative AI

The authors used Claude in order to: Spell check.

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

## A. Graph Schema

```
@prefix : <http://hiddencostreport.org/schema#> .
@prefix rdf: <http://www.w3.org/1999/02/22-rdf-syntax-ns#> .
@prefix rdfs: <http://www.w3.org/2000/01/rdf-schema#> .
@prefix owl: <http://www.w3.org/2002/07/owl#> .

:Company a rdfs:Class .
:Product a rdfs:Class .
:Metric a rdfs:Class .
:Observation a rdfs:Class .

:Name a rdf:Property ;
  rdfs:range rdf:Literal .

:OpenCorporatesID a rdf:Property ;
  rdfs:domain :Company ;
  rdfs:range rdf:Literal .

:hasMetric a rdf:Property;
  rdfs:domain :Company ;
  rdfs:range :Observation .

:MetricTitle
  rdfs:domain :Metric ;
  rdfs:range rdf:Literal .

:MetricDesigner
  rdfs:domain :Metric ;
  rdfs:range rdf:Literal .

:Questions
  rdfs:domain :Metric ;
  rdfs:range rdf:Literal .

:ValueType
  rdfs:domain :Metric ;
  rdfs:range rdf:Literal .

:Unit a rdf:Property ;
  rdfs:domain :Metric ;
  rdfs:range rdf:Literal .

:Brand
  rdfs:domain :Product .

:Ingredient
  rdfs:domain :Product .

:ManufacturedIn a rdf:Property ;
  rdfs:domain :Product .
```

# B. Truepricemethod Cost Table

*Source: https://github.com/Truepricemethod/Monetisation_factors, CC BY 4.0*

| indicator | factor | unit |
|---|---:|---|
| airp_acid | 5.47 | EUR/kg SO2 eq |
| airp_ozone | 70.4 | EUR/kg CFC-11eq |
| airp_pm | 81.3 | EUR/kg PM2.5-eq |
| airp_smog_eco | 3.33 | EUR/kg NOx-eq |
| airp_smog_hh | 0.118 | EUR/kg NOx-eq |
| airp_tox_freshwater | 0.0472 | EUR/kg 1,4-DB eq |
| airp_tox_human | 129,000 | EUR/DALY |
| airp_tox_marine | 0.00215 | EUR/kg 1,4-DB eq |
| airp_tox_terrestrial | 0.000294 | EUR/kg 1,4-DB eq emitted to industrial soil |
| climate | 0.312 | EUR/kgCO2eq |
| fossil | 0.560 | EUR/kg oil eq |
| land_occupation_coastalwetland | 12,800 | EUR/(ha*yr) |
| land_occupation_grassland | 2,830 | EUR/(ha*yr) |
| land_occupation_inlandwetland | 17,400 | EUR/(ha*yr) |
| land_occupation_otherforest | 1,180 | EUR/(ha*yr) |
| land_occupation_tropicalforest | 2,470 | EUR/(ha*yr) |
| land_occupation_woodland | 1,600 | EUR/(ha*yr) |
| land_trans_coastalwetland | 3,770 | EUR/ha |
| land_trans_grassland | 340 | EUR/ha |
| land_trans_inlandwetland | 43,100 | EUR/ha |
| land_trans_otherforest | 3,120 | EUR/ha |
| land_trans_tropicalforest | 4,510 | EUR/ha |
| land_trans_woodland | 1,290 | EUR/ha |
| material | 0.283 | EUR/kg Cu eq |
| wateruse | 1.62 | EUR/m3 |
| soil_compaction | 0.64 | EUR/tkm |
| soildeg_carbon | 0.0353 | EUR/kg SOC loss |
| soildeg_watererosion | 0.0268 | EUR/kg soil loss |
| soildeg_winderosion | 0.0343 | EUR/kg soil loss |
| soilp_tox_freshwater | 0.0472 | EUR/kg 1,4-DB eq |
| soilp_tox_human | 129,000.0000 | EUR/DALY |
| soilp_tox_marine | 0.00215 | EUR/kg 1,4-DB eq |
| soilp_tox_terrestrial | 0.000294 | EUR/kg 1,4-DB eq emitted to industrial soil |
| waterp_eu_fresh | 239 | EUR/kg P eq to freshwater-risk adjusted |
| waterp_eu_marine | 16.6 | EUR/kg N eq to marine water |
| waterp_tox_freshwater | 0.0472 | EUR/kg 1,4-DB eq |
| waterp_tox_human | 129,000 | EUR/DALY |
| waterp_tox_marine | 0.00215 | EUR/kg 1,4-DB eq |
| waterp_tox_terrestrial | 0.000294 | EUR/kg 1,4-DB eq emitted to industrial soil |
| child_audit | 8.75 | EUR/FTE |
| cl_haz | 42.0 | EUR/hour of hazardous child labour |
| cl_non_haz | 15.3 | EUR/hour of non-hazardous child labour |
| ot_audit | 8.75 | EUR/FTE |
| ot_illegal | 125 | EUR/FTE |
| ot_underpaid | 125 | EUR/FTE |
| ot_wagegap | 1.03 | EUR/EUR |

| | | |
|---|---:|---|
| fl_abuse | 43,400 | EUR/FTE |
| fl_audit | 8.75 | EUR/FTE |
| fl_debt | 20,600 | EUR/FTE |
| fl_workers_high | 139,000 | EUR/FTE |
| fl_workers_low | 14,000 | EUR/FTE |
| fl_workers_med | 76,600 | EUR/FTE |
| dis_gender_audit | 8.75 | EUR/FTE |
| dis_gender_eqgap | 1.03 | EUR/EUR |
| dis_gender_mlgap | 1.03 | EUR/EUR |
| dis_gender_mlworkers | 2,000 | EUR/FTE |
| dis_gender_wagegap | 1.03 | EUR/EUR |
| income | 1.03 | EUR/EUR |
| dfa | 430 | EUR/violations |
| dfa_audit | 8.75 | EUR/FTE |
| ss_audit | 8.75 | EUR/FTE |
| ss_plgap | 1.03 | EUR/EUR |
| ss_workers | 2,650 | EUR/FTE |
| ohs_audit | 8.75 | EUR/FTE |
| ohs_breachfte | 2,160 | EUR/FTE |
| ohs_breachinj | 4,790 | EUR/incidents |
| ohs_f | 3,840,000 | EUR/incidents |
| ohs_nf_ins | 4,520 | EUR/incidents |
| ohs_nf_nins | 4,670 | EUR/incidents |
| har_audit | 8.75 | EUR/FTE |
| har_np_ns | 27,800 | EUR/worker |
| har_np_s | 27,800 | EUR/worker |
| har_p_ns | 68,500 | EUR/worker |
| har_p_s | 76,800 | EUR/worker |
| har_p_ss | 86,100 | EUR/worker |
| wage_audit | 8.75 | EUR/FTE |
| wage_gap_min | 1.53 | EUR/EUR |
| wage_gap_minlw | 1.03 | EUR/EUR |