# OpenReview forum: "Open Data Fusion for Approximate True Cost Accounting using Knowledge Graphs"
_ijcai.org/IJCAI-ECAI/2026/Workshop/GENAIK-NORA — IJCAI-ECAI 2026 Joint Workshop on GENAIK and NORA_

### Official Review · Reviewer_nxVm · 2026-05-25
**Promising KG-based demo for approximate true cost accounting, but evaluation and methodological grounding are limited**

**Rating:** 6
**Confidence:** 3

**Review:**

This paper presents a demo system for Approximate True Cost Accounting (ATCA) using a knowledge graph constructed from open data sources, primarily Wikirate and OpenProductsFacts. The system aims to estimate hidden externality costs for companies and products by harmonizing ESG-related metrics, mapping some of them to monetization factors, and exposing the resulting graph through a UI, QLever endpoint, and text-to-SPARQL interface.

Overall, I find the problem important and well-suited to a workshop on knowledge graphs and agentic systems. The paper makes a reasonable case that true cost accounting is difficult to scale manually, and that open-data fusion through a KG could make such estimates more accessible. The public demo and open-source code are also valuable contributions. However, the current submission is much stronger as a position/demo paper than as a mature research contribution. The methodology is based on very strong simplifying assumptions, and the paper provides little quantitative evaluation of data quality, approximation error, KG construction quality, metric categorization, or text-to-SPARQL performance.

The paper would be significantly stronger with a more concrete evaluation section. Even a demo paper could include:

1. KG statistics: triples, entities, observations, products, companies, categories, and coverage.
2. Categorization evaluation on a manually labeled sample of metrics.
3. A small benchmark against existing true-price reports, even if only 3–5 products.
4. Error decomposition showing how missing data, categorization mistakes, and allocation assumptions affect final estimates.
5. A clearer schema with provenance, time, location, units, and uncertainty.
6. A description and evaluation of the text-to-SPARQL component.
7. Correction of the numerical inconsistency in the Bocca Coffee example.

I see this as a promising and relevant demo/position paper with a useful public artifact. However, the scientific claims are currently limited by weak evaluation, strong assumptions, and insufficient detail about the core data-fusion and monetization steps. For a workshop demo track, I lean mildly positive because the system is open, relevant, and likely to stimulate useful discussion. For a full research track, I would be more critical.

---

### Official Review · Reviewer_jaCL · 2026-06-04
**System for true price that uses a knowledge graph.**

**Rating:** 5
**Confidence:** 4

**Review:**

The paper describes a system that uses a knowledge graph (KG) for data for estimating a price that also reflects externalities (the "true price"). It is an interesting topic and the paper points to relevant resources I was not aware of.

The paper is only partially relevant for the workshop. Indeed the work around the KG is described, but the interface between KGs vs. generative AI and agentic systems is hardly treated. A text-to-SPARQL facility within the system is mentioned, but no technical details about this aspect is given other than it is zero-shot LLM-aided and error-prone.

In appendix B, there is a large list of cost. This seems to be just a copy data available from https://github.com/Truepricemethod/Monetisation_factors under CC BY license.

The are a number of unclear aspects with the manuscript.

An example is given with computation of a true price for "Bocca Coffee". The paper reference a Web page https://www.trueprice.org/projects/bocca-coffee/ where there is a true price calculation for per cup (10 g) and per kilogram of coffee. The paper gives the true price to 9.21 euro per kilogram but the referenced web paper reports 3.71 euro per 10 gram. I cannot resolve this inconsistency.

Somewhat confusingly, the price is given with three digits after the decimal point.

During data ingestion to the knowledge graph a step referred to as "hamonization/harmonized" is mentioned but it is not described what it is. Some form of processing to align the items Wikidata and OpenFacts resources?

In the abstract, it is claimed that "Reflecting negative externalities such as pollution in market prices is necessary
for long-term price stability." I am wondering whether this is really true? I think the sentence should be removed or supported by a reference.

The abstract mentions that the system described in the paper is available on a web page https://atca.health-nlp.com/ I am not able to access that web page.

The system uses bout Oxigraph and Qlever. The choice of Qlever is argued to be due to autocompletion and faster string search, but then why is Oxigraph used? It seems redundant to use two triple stores.

---

### Official Review · Reviewer_hTsF · 2026-06-05
**Review: Open Data Fusion for Approximate True Cost Accounting using Knowledge Graphs**

**Rating:** 6
**Confidence:** 4

**Review:**

## Summary

The paper presents a demo system that approximates True Cost Accounting (TCA) for companies and products by fusing data from Wikirate (ESG metrics) and OpenProductsFacts into an RDF knowledge graph, then applying a published monetization table (True Price Foundation) to categorized metrics. The system is exposed through a UI, a QLever SPARQL endpoint, and an LLM-based text-to-SPARQL interface. The authors demonstrate the approach on Bocca Coffee, comparing against a published true-price report. The KG covers ~1000 companies and ~1000 metrics; the live demo is at atca.health-nlp.com with code on GitHub.

## Strengths

1. Important problem, underserved space. Automating externality pricing is a genuinely valuable direction. Existing efforts (True Price Foundation, TEEBAgriFood) are bespoke, agriculture-skewed, and slow. A reusable, industry-agnostic pipeline that ingests public data is a useful contribution even in approximate form.
2. Working demo and open artifacts. The live system, open-source code, schema, and reproducibility of the worked example are commendable for a demo paper. The QLever endpoint and federated-query design (with the Wikidata example) lower the barrier for downstream reuse.
3. Sensible architecture. Separating ingestion, storage, and uses; using Oxigraph for storage and QLever for indexing/autocomplete; supporting federation via OpenCorporatesID — these are good engineering choices and the schema (Appendix A) is reasonable.
Honest treatment of limitations. Section 3 explicitly names the heroic assumptions (proportional contribution, vertical integration, location-independent costs) and the brittleness of the heuristic harmonization. This candor is welcome.
4. Workshop fit. The combination of KG construction, federated SPARQL, and an LLM-based text-to-SPARQL agent maps directly to the NORA workshop's KG-and-agent theme.

## Weaknesses
1. The central methodology reduces to a linear rescaling of price. The assumption that "all products contribute to all externalities in proportion to their price" means the approximate true price is, in the per-product case, just the market price multiplied by (1 + hidden cost per euro of revenue). This makes the output less informative than the framing suggests: relative product comparisons within a company become trivial, and cross-company comparisons depend almost entirely on the revenue normalization. The paper should confront this more directly — perhaps by showing per-company hidden-cost-per-euro distributions across the 900 companies and discussing what the number actually buys the reader beyond a company-level ESG score.
2. Evaluation is a single anecdote. Figure 1 shows one company (Bocca Coffee) where the approximation reaches 8.38€/kg against a ground truth of 9.32€/kg. From this one point, the paper concludes the approach "undershoots due to incomplete data." This is plausible but unsupported. Even given the acknowledged absence of ground-truth corpora, the authors could: (a) report the comparison across the handful of published true-price studies they cite ([3]–[6]); (b) compute internal consistency checks (e.g., how stable estimates are when one metric is removed); (c) report coverage statistics (what fraction of the 900 companies have enough categorized metrics to produce a non-trivial estimate). A demo paper does not need a full benchmark, but a single data point is too thin to support the central empirical claim.
3. Categorization coverage is low and the explanation is hand-waved. Only ~30% of metrics get assigned a category by the keyword heuristic. The claim that this is "close to the ceiling" because metrics like "number of employees" are not directly costable deserves evidence — for instance, a manual sample of the uncategorized 70% labelled as "categorizable but missed," "ambiguous," or "genuinely non-monetizable." Without this, the ceiling claim is speculation, and the reader cannot tell whether the bottleneck is the heuristic or the data.
4. The "reduction of hazardous waste" miscategorization example points to a deeper problem. A keyword matcher cannot distinguish a metric that measures an externality from one that measures reduction of it, yet these have opposite signs in the cost computation. This issue likely systematically biases estimates and warrants quantification rather than mention in passing.
5. Text-to-SPARQL is presented as a feature but described as error-prone. The text-to-SPARQL interface is named in the keywords and listed as a data use, but the paper says only that it is zero-shot and "error-prone." Either evaluate it (even a small held-out set of natural-language queries with execution accuracy would suffice for a demo) or remove the prominent positioning. As written, this component lowers rather than raises confidence in the demo.
6. Schema appendix appears incomplete. In Appendix A, properties :MetricTitle, :MetricDesigner, :Questions, :ValueType, :Brand, and :Ingredient are listed without a rdf:Property declarations or domain/range — they appear to be placeholders or formatting errors. Please clean up before publication.
7. EEIOA dismissed too quickly. Davis & Caldeira and Hong et al. are characterized as "avoiding the step of converting consumption to cost equivalent." That is fair, but EEIOA methods would compose naturally with the monetization table used here: input-output tables give per-sector externality intensities at much better coverage than crowdsourced ESG reports. A paragraph on why the authors chose company-level open data over a hybrid with sector-level EEIOA would strengthen the related-work positioning.
8. The "true price" framing risks overclaiming. Because only categorized metrics contribute, the output is structurally a lower bound on a subset of externalities, not an approximation of the true price. The Bocca example's directional bias (undershooting) is consistent with this. Reframing the deliverable as "an open lower-bound estimator" or "an externality-cost floor" would be more defensible and would not weaken the practical value.

## Minor Issues

1. "Figure 1" caption: the comparison would benefit from a brief note in the caption (or near it) on what fraction of the gap comes from missing categories vs. missing data vs. the proportionality assumption.
2. The QLever and Oxigraph choices are mentioned but never compared or justified against alternatives (Virtuoso, GraphDB, Blazegraph) — a sentence on why this stack would help reproducibility.
3. The federated-query example (Figure 2) retrieves a company motto, which is not relevant to TCA. Pick an example that exercises the actual use case (e.g., pulling sector codes or geolocation to refine cost lookups).
4. "atca.health-nlp.com" — the health-nlp subdomain is surprising given the topic; a one-line note on the hosting context would prevent reader confusion.
5. Several minor wording issues ("our will" → "ours will" in Section 3; "monetisation"/"monetization" inconsistency).

---

### Official Review · Reviewer_KH7w · 2026-06-06
**A simple approach to building knowledge graphs for true cost accounting**

**Rating:** 6
**Confidence:** 3

**Review:**

The authors of this submission build a knowledge graph (KG) for true cost accounting (TCA), a technique that estimates externalities in products not reflected by their price. In particular, the KG is constructed for the agrifood sector, after ingesting three different datasets  and is exposedvia three endpoints (user interface, Qlever and text2sparql). The authors also provide a live demo of their approach (the website didn't work, though, due to Server Name Identification issues) along with a Github repository for their code. Overall, the submission feels more like a student project, yet the workshop audience may benefit from an applied KG creation approach in the fields of economy and cost estimation, hence my weak accept evaluation.

---

### Decision · Program_Chairs · 2026-06-10

Accept